# Approaches to Reduce Rice Blast Disease Using Knowledge from Host Resistance and Pathogen Pathogenicity

**DOI:** 10.3390/ijms24054985

**Published:** 2023-03-05

**Authors:** Muhammad Usama Younas, Guanda Wang, Haibo Du, Yi Zhang, Irshad Ahmad, Nimra Rajput, Mingyou Li, Zhiming Feng, Keming Hu, Nasr Ullah Khan, Wenya Xie, Muhammad Qasim, Zongxiang Chen, Shimin Zuo

**Affiliations:** 1Key Laboratory of Plant Functional Genomics of the Ministry of Education/Jiangsu Key Laboratory of Crop Genomics and Molecular Breeding, Agricultural College of Yangzhou University, Yangzhou 225009, China; 2Co-Innovation Center for Modern Production Technology of Grain Crops of Jiangsu Province, Key Laboratory of Crop Genetics and Physiology of Jiangsu Province, Yangzhou University, Yangzhou 225009, China; 3Department of Plant Breeding & Genetics, Faculty of Agriculture, Gomal University, Dera Ismail Khan 29111, Pakistan; 4Joint International Research Laboratory of Agriculture and Agri-Product Safety, The Ministry of Education of China, Institutes of Agricultural Science and Technology Development, Yangzhou University, Yangzhou 225009, China

**Keywords:** *Oryza sativa*, blast disease, resistance gene, quantitative trait locus, avirulence gene

## Abstract

Rice is one of the staple foods for the majority of the global population that depends directly or indirectly on it. The yield of this important crop is constantly challenged by various biotic stresses. Rice blast, caused by *Magnaporthe oryzae* (*M. oryzae*), is a devastating rice disease causing severe yield losses annually and threatening rice production globally. The development of a resistant variety is one of the most effective and economical approaches to control rice blast. Researchers in the past few decades have witnessed the characterization of several qualitative resistance (*R*) and quantitative resistance (*qR*) genes to blast disease as well as several avirulence (*Avr*) genes from the pathogen. These provide great help for either breeders to develop a resistant variety or pathologists to monitor the dynamics of pathogenic isolates, and ultimately to control the disease. Here, we summarize the current status of the isolation of *R*, *qR* and *Avr* genes in the rice–*M. oryzae* interaction system, and review the progresses and problems of these genes utilized in practice for reducing rice blast disease. Research perspectives towards better managing blast disease by developing a broad-spectrum and durable blast resistance variety and new fungicides are also discussed.

## 1. Introduction

Rice (*Oryza sativa*) is pivotal to human life, especially in Asia where millions of people depend either directly or indirectly on rice consumption for calorie uptake. Being a major cereal crop and critical to global food security, the United Nations Organization (UNO) has declared 2004 as the International Year of Rice [1]. The growth, development and yield of rice is constantly challenged by various biotic and abiotic stresses. Abiotic stresses include drought, salinity and nutrient deficiency while biotic stresses challenge rice production in the form of insect attack and viral, bacterial and fungal diseases. Among these biotic stresses, a fungal disease called rice blast poses a great threat to rice yield, which is caused by *Magnaporthe oryzae* (*M. oryzae*); it accounts for yield losses up to 10–30% annually, and complete loss (100%) in the years of pandemics [2,3,4]. Currently, rice blast disease is prevalent in more than 85 countries [5]. Several factors such as high humidity (>80%), cloudy and wet weather, low temperature (15–25%) and excessive application of fertilizers may increase the incidence of rice blast [6,7]. *M. oryzae* may infect aerial parts of the rice plant at any development stage, which causes seedling blast, leaf blast, node blast, grain blast and neck blast, based on the appearance of lesions on different rice parts [8]. The pathogen spreads in rice biotrophically during early infection and then switches to the necrotrophic phase in 4 to 5 days, which is considered a hemibiotrophic pathogen. In the biotrophic stage, the blast fungus produces a unique cell called an appressorium, which is a prerequisite for its infection [9]. 

Chemicals, fungicides especially, are the easiest and most general method to control rice blast. The most successful and effective fungicides for the control of rice blast in Japan are copper fungicides [10]. However, it soon became apparent that copper fungicides are phytotoxic and adversely affect both human health and soil microbiota. Then, a mixture of phenylemercuric acetate (PMA) and copper fungicides proved effective agents compared to copper alone to manage rice blast with less toxicity to both human health and the rice plant. With the extensive application of one kind of fungicide, the emergence of resistance to the fungicide in *M. oryzae* was reported. As a result, the application of different fungicides or a combination thereof in rotation has been widely recommended in practice to manage rice blast. Currently, lots of different kinds of fungicides have been developed to be used solely or in combination to curb rice blast disease [11]. However, despite widespread applications in plant pathogen control and plant protection, fungicides pose a great threat to human health and the environment. Fungicides have been shown to drastically decline beneficial microorganisms and the fungal community in soil [12,13]. Some of the microbes are key to soil texture transformation, nitrogen fixation, nutrient use efficiency, decomposition of organic matter, soil structure and fertility [12,13,14,15,16]. 

It is well known that developing resistant varieties with resistance genes is the most effective and economical strategy to control the disease. There has been a focus in the past few decades to identify such resistance genes that can combat *M. oryzae* strains and transfer them through breeding into susceptible cultivars; thus, such resistant cultivars can be used as safer alternatives to toxic fungicides. Such classical breeding approaches are complemented by the combination of classical breeding and genetic engineering to develop broad-spectrum resistance cultivars [17]. With the rapid development of molecular biology and genetics, both qualitative resistance (*R*) and quantitative resistance (*qR*) genes to rice blast have been extensively identified from rice natural varieties as well as wild rice. Despite a few *R* genes showing broad-spectrum resistance to multiple strains/isolates, most of them presented a typical “gene-for-gene” resistance to some strains/isolates. Corresponding to some *R* genes, a few avirulence (*Avr*) genes have been isolated from different strains/isolates. When the host R protein recognizes the pathogen Avr protein, resistance is activated, while the sequence variation of either one will lead to host susceptibility. Therefore, knowledge about *Avr* genes and their interaction mechanism with the corresponding *R* genes could be used to guide disease management. With respect to rice breeding, broad-spectrum resistance is highly preferred in agriculture practice. Therefore, how to develop a broad-spectrum and durable resistance variety is particularly important for rice production, as well as a need to comprehensively understand both *R* genes from the rice host and *Avr* genes from *M. oryzae.*


In this review, we summarize the current status of isolated *R* and *qR* genes from rice and *Avr* genes from *M. oryzae*, and the progresses of these genes that have been utilized in rice breeding and disease management. 

## 2. Isolation and Characterization of Qualitative and Quantitative Resistance Genes to Rice Blast

The rice genome has a repertoire of resistance genes, some of which confer race-specific resistance and others which confer broad-spectrum resistance. To date, more than 100 *R* genes have been identified and mapped to different chromosomes of the rice genome and, among them, up to 38 *R* genes scattered on all rice chromosomes have been cloned (Table 1). Among 12 chromosomes, chromosome 11 carries the highest number (28) of *R* genes, while chromosomes 3, 7 and 10 possess a single *R* gene. The majority of the identified *R* genes are located in clusters, generally in three major clusters on chromosomes 6, 11 and 12. For instance, the cluster on chromosome 6 contains at least 7 *R* genes, *Piz-t*, *Pi9*, *Pigm*, *Pi2*, *Pi40*, *Piz* and *Pi54*, which all encode proteins with the nucleotide-binding domain (NB-domain) and leucine-rich repeats (LRR) domain. Wide variations on either coding regions or copy numbers of NBS-LRR genes in this cluster were reported. The *Pigm* donor variety ‘‘Gumei4′’ contains the highest number (13) of NBS-LRR genes compared with the donors of other *R* genes in this cluster. The presence of 18 aa changes in the LRR domain of *Piz-t* and not only specifies the resistance induction but also renders *Piz-t* distinct from *Pi2* [18]. Although the majority of *R* genes encode NBS-LRR proteins, some are exceptions such as *Pi-d2*, *pi21* and *Ptr*. The *pi21*, cloned from rice cultivar ‘Owarihatamochi’, encodes a protein rich in proline amino acids [19]; *Pi-d2,* isolated from ‘Digu’, encodes the aB-lectin receptor kinase [20]; *Ptr* on chromosome 12 from M2354 encodes an ARM repeat domain protein [21]. *Pi21*, *Pi5*, *Pi63* and *Pb1* are pathogen-inducible expressions, while almost all the remaining *R* genes are constitutively expressed no matter the existence or not of *M. oryzae* [22]. Most of the cloned *R* genes confer resistance to rice blast pathogen at the seedling stage, while others such as *Pi68*, *Pi25*, *Pb1* and *Pi64* show resistance at the seedling as well as the adult stage [23,24,25,26]. Although most of these *R* genes show race-specific resistance, some *R* genes, such as *Piz-t*, *Pi9*, *Pigm*, *Pi2* and *Pi54,* located at chromosome 11, were reported to confer broad-spectrum resistance, especially for *Pi9*, *Piz-t* and *Pigm* [18,27].

With respect to a resistance mechanism, plants have evolved a complex system to recognize and respond to pathogen attack with the help of specialized receptors known as pattern-recognition receptors (PRRs). Such a primary line defense to halt pathogens is termed pattern-triggered immunity (PTI) [28,29]. PTI not only acts to curtail pathogen invasion but also to maintain normal microbiota inside the plant leaf which is beneficial for plant health [28,29,30,31]. However, in order to infect successfully, pathogens may further secrete various types of virulence-causing molecules, also commonly known as effectors, and deliver them inside the plant cell or apoplast in order to bypass PTI. To resist the infection, plants have evolved a series of special class intracellular receptors to recognize effectors and trigger a second line of immunity, commonly known as effector-triggered immunity (ETI), to limit the spread of the pathogen. In general, these intracellular receptors are very conserved with the typical domain of NBS-LRR, also called NBS-LRR receptors (NLRs), which have been widely identified in plants against various pathogens [32]. This ‘zig-zag’ model under the mechanism of PTI–ETI interplay was proposed by Dangl in 2006, who suggested it was a two-layer defense system to respond to different types of pathogens. However, how PTI and ETI interplay contributes to qualitative or quantitative resistance in plants has remained a hot topic of debate in the scientific community in the past decade [33]. Extensive research on this paradigm concluded that the activation of two distinct classes of receptors, PRRs and NLRs, during PTI and ETI leads to cascades of early signaling that ultimately defeat the pathogen [33,34,35]. The signaling cascades are manifested in different outputs, such as ROS, calcium flux, hormone signaling, transcription reprogramming and mitogen-activated protein kinase (MAPK) cascade [36,37,38]. These outputs pinpoint the intersectional points of PTI–ETI interplay to ensure robust immunity against a plethora of pathogens. 

The majority of the cloned *R* genes for rice blast encoding NBS-LRRs proteins follow a similar mechanism of action as ETI, represented by the hypersensitive response (HR) with the phenomena of ROS burst and programmed cell death (PCD). The host elicitor proteins are recognized directly or indirectly by host NLR proteins in cytoplasm. Upon recognition, a cascade of downstream signaling pathways is directed that ultimately combats the pathogen elicitors. One of the classical examples of blast *R* genes is *Pigm*, which is a multi-allelic locus and encodes clusters of NBS-LRRs including cognate pairs of receptors such as *PigmS* and *PigmR*. The former is a weak attenuator of pathogen pathogenicity via homodimerization while the latter confers broad-spectrum resistance to multiple races of rice blast. Moreover, both of these pair sustains a balance between yield and resistance against blast [39]. Additional aspects of the mechanistic study of *Pigm* unraveled an additional partner of *Pigm* known as PIBP1 (PigmR-Interacting and Blast Resistance Protein 1), which encodes an RNA recognition motif (RRM)-containing protein. PIBP1 is found to interact only with NLR proteins of rice blast resistance genes conferring broad-spectrum resistance [40]. PIBP1 is a transcriptional factor that directly activates other transcription factors such as OsPAL1 and OsWAK14 that ultimately lead to ETI-based broad-spectrum resistance [27]. In fact, almost all *R* genes-mediated immunity may produce a typical ETI response, but the signals linking *R* genes and ETI, represented by the phenomena of ROS burst and PCD, remain largely unknown.

Different studies have demonstrated that rice resistance to *M. oryzae* is complex and often involves the interaction of various *qR* and *R* genes in a synergistic approach [41]. Contrary to the *R* gene, *qR* genes, such as *Pi21*, *bsr-d1* and *bsr-k1*, generally deploy a different type of resistance mechanism against rice blast. One such classic example is *Pi21*, which encodes a protein rich in proline amino acids and carries a protein–protein interacting domain and metal-binding domain. The presence of the heavy metal domain suggests that *Pi21*’s inherent capability of metal transport might be associated with broad-spectrum resistance. A 1705 bp long polymorphic region in the ORFs of resistant and susceptible alleles of *Pi21* was identified, and the resistant allele had undergone two key mutations (21 and 48 bp) in the proline-rich region. These two mutations in the key polymorphic region of resistance rice lines hinder the access of this target region being targeted by the product of *pi21* negative regulators, thus contributing to the broad-spectrum resistance of *Pi21* against rice blast [42]. Another example is *bsr-d1,* which was identified from a well-known rice cultivar ‘Digu’ with broad and durable blast resistance [43]. The *bsr-d1* encodes a C2H2 transcription factor (TF), and one critical SNP variation in its promoter region leads to an upstream TF MYBS1 that is more strongly binding to the promoter of *bsr-d1* in response to the *M. oryzae* infection and then suppresses its transcription. Since BSR-D1 protein is one of the directly positive regulators of peroxidase encoding genes, functioning in decreased H_2_O_2_, the low level of BSR-D1 in *bsr-d1* plant results in it accumulating more H_2_O_2_ and ultimately limiting the spread of *M. oryzae*. H_2_O_2_ plays a great part in plant immune response and *MABS1* is a commonly inducible gene by *M. oryzae*, which accounts for the broad-spectrum resistance of *bsr-d1* to blast.

**Table 1 ijms-24-04985-t001:** Information about cloned rice blast resistance genes.

Gene/Allele	Mapped Position	Protein Encoded	Cognate *Avr* Genes	Donor	References
*Pi-b*	Chr. 2	NLR	*Avr-Pib*	Tohuku IL9	[44]
*Pit*	Chr. 1	NLR	Unknown	K59	[45]
*Pish*	Chr. 1	NLR	Unknown	Nipponbare	[46]
*Pi35*	Chr. 1	NLR	Unknown	Hokkai 188	[47]
*pi21*	Chr. 4	Proline-rich metal binding protein	Unknown	Owarihatamochi	[42]
*Pi37*	Chr. 1	NLR	Unknown	St. No. 1	[48]
*Pi64*	Chr. 1	NLR	Unknown	Yangmaogu	[26]
*Pi63*	Chr. 4	NLR	Unknown	Kahei	[49]
*Pi2*	Chr. 6	NLR	Unknown	Jefferson	[18]
*PiPR1*	Chr. 4	NLR	Unknown	Unknown	[50]
*Pi9*	Chr. 6	NLR	*AVR-Pi9*	75-1-127	[51]
*Piz-t*	Chr. 6	NLR	*Avr-Pizt*	Zenith	[18]
*Pizh*	Chr. 6	NLR	Unknown	Unknown	[52]
*Pigm*	Chr. 6	NLR	Unknown	Gumei4	[27]
*Pi50*	Chr. 6	NLR	Unknown	Er-Ba-zhan	[52]
*Pi-d2*	Chr. 6	B-lectin receptor Kinase	Unknown	Digu	[20]
*Ptr*	Chr. 12	ARM repeat domain protein	Unknown	M2354	[21]
*Pi-d3*	Chr. 6	NLR	Unknown	Digu	[53]
*Pi36*	Chr. 8	NLR	Unknown	Kasalath	[54]
*Pid3-A4*	Chr. 6	NLR	Unknown	*Oryzarufipogon*	[55]
*Pi25*	Chr. 6	NLR	Unknown	Gumei2	[23]
*Pi56*	Chr. 9	NLR	Unknown	Sanhuangzha	[56]
*Pb1*	Chr. 11	NLR	Unknown	Modan	[57]
*Pi5*	Chr. 9	NLR	Unknown	RIL260	[58]
*Pii*	Chr. 9	NLR	AVR-Pii	Hitomebore	[59]
*Pike*	Chr. 11	NLR	Unknown	Xiangzao143	[60]
*Pi1*	Chr. 11	NLR	Unknown	C101LAC	[61]
*Pik-h*	Chr. 11	NLR	Unknown	K3	[62]
*Pikm*	Chr. 11	NLR	*AVR Pikm*	Tsuyuake	[63]
*Pik-p*	Chr. 11	NLR	*AVR-Pikp*	K60	[64]
*Pik*	Chr. 11	NLR	*AVR-Pik*	Kusabue	[56]
*Pi-ta*	Chr. 12	NLR	*AVR-Pita*	Yashiro-mochi	[65]
*Pi65*	Chr. 12	LRR-RLK	Unknown	GangYu129	[66]
*Pi-CO39*	Chr. 11	NLR	AVR-CO39	CO39	[67]
*Pia*	Chr. 11	NLR	*AVR-Pia*	Sasanishiki	[68]
*Pi54rh*	Chr. 11	NLR	*Avr-Pi54*	*O. rhizomatis*	[69]
*Pi54*	Chr. 11	NLR	*AVR-Pi54*	*O. officinalis*	[70]
*Pi54*	Chr. 11	NLR	*AVR-Pi54*	Tetep	[71]

## 3. Isolation of Avirulence Genes from Rice Blast Pathogen

At the moment, more than 26 *Avr* genes have been mapped in the *M. oryzae* genome and 14 of them have already been cloned (Table 2). With the exception of *ACE1* and *AVR-Pita*, most of these *Avr* genes code for secretory proteins with less than 200 amino acids [72]. ACE1 is a non-secretory protein of secondary metabolite origin that exists as a hybrid of non-ribosomal peptide synthetase (NRPS) and polyketide synthase (PKS). The NRPS part containing the β-ketoacyl synthase domain has been shown to elicit avirulence. Fifteen of the 24 *Avr* genes mapped so far have been located near the chromosomal ends, while five *Avr* genes are interspersed by transposons on either one or both sides of the *Avr* genes. The existence of transposons on either side of the *Avr* genes supports the hypothesis that *Avr* genes might have experienced a gain or loss of function during pathogen evolution. Besides, nine cloned genes have presence/absence polymorphism in the rice-infecting population. The predominant *Avr* gene *AVR-Pia* is known to have been acquired by different isolates or at least translocated between chromosomes 5 to 7 in different isolates of *M. oryzae*. 

With respect to *R-Avr* interactions, several interaction modes were observed based on the advances of seven *R-Avr* pairs. Two of these pairs, *Pii/AVR-Pii* and *Piz-t/AvrPiz-t*, interact indirectly and recognize each other. The remaining five pairs interact directly in three different ways. The first way is a classical gene-for-gene model, one Avr protein of the pathogen is directly recognized by a corresponding R protein, for instance, *Pi54/AVR-Pi54* and *Pi-ta/AVR-Pita* [72]. One of the earliest studied *R*–*Avr* interactions is the *Pi-ta/AVR-Pita* pair in *M. oryzae* that laid the foundation for plant–pathogen interaction and their interplay in the onset of disease or development of resistance [73]. *AVR-Pita*, a telomere like the *Avr* gene, encodes a secreted protein with a distinct Zn-metalloprotease domain. The mature form of *Avr-Pita* is protease containing 176 aa at the C-terminus [74]. *Avr-Pita* belongs to a special class of the *AVR-Pita* gene family with three distinct genes, i.e., *AVR-Pita1*, *AVR-Pita2* and *AVR-Pita3*. The former two are functional genes triggering *Pita*-mediated resistance while the latter one is a pseudogene without *Avr* function [75]. The corresponding *Pita* gene of the pair is a classical NLR (928 aa) receptor localized in the cytoplasm and expressed constitutively [65]. The leucine-rich domain (LRD) of the Pita protein directly interacts with the AVR-Pita_176_ protein and induces downstream signaling cascades. Functional validation through site-directed mutagenesis has shown that the AVR-Pita lose avirulence function by substituting two amino acids, i.e., avr-pita_176_^E177D^ and avr-pita_176_^M178W^. Similarly, a mutant of the *Pita* gene with a single amino acid substitution (LRD^A918S^) diminishes the AVR-Pita_176_ -Pita LRD physical interaction, suggesting the practical outcomes of *R*–*Avr* pair interplay in the development of immunity against *M. oryzae* [76]. Besides the interaction between Pita–AVR-Pita, Han et al. (2021) recently found that Avr-Pita was able to interact with a cytochrome *C* oxidase (COX) assembly protein, OsCOX11, in mitochondria to reduce ROS accumulation for suppressing rice innate immunity [73]. The second way is that one Avr protein is sensed, interacts with two R protein homologs, and triggers an immune response upon recognition [56,65,77,78]. *Pia* locus comprises two NLRs proteins called RGA4 and RGA5, oriented face to face in opposite directions. Both proteins interact with a single Avr protein AVR-Pia with an N-terminal secretory protein [79]. Isolates of the *M. oryzae* avirulent to the *Pia* gene in rice contain 1-3 copies of *AVR-Pia*, depending on the specific isolate [80]. RGA5 alternative splicing produces two isoforms, RGA5-A and RGA5-B, of which only RGA5-A mediates *Pia* resistance. The constitutive expression of RGA4 causes cell death, which is then prevented by RGA5 in planta in the absence of infection, according to in vitro experiments. The NB domain of RGA4 is mandatory for cell death induction [80,81,82]. The physical interaction of AVR-Pia with the C-terminal non-LRR domain of RGA5 relieves the inhibition status and stimulates RGA4-mediated cell death. The third interaction works as a decoy model in which resistance mediated by the *R-Avr* pair is determined by an additional decoy protein interacting with the *R-Avr* pair. The best example is the *Pii/Avr-pii* pair interacting with two additional rice proteins, OsExo70-F2 and OsExo70-F3. The simultaneous knockdown of OsExo70-F2 and OsExo70-F3 completely diminished *Pii* immune receptor-dependent resistance against *Avr-pii.* The interaction of OsExo70-F3 with pathogens *AVR-Pii* is mandatory to induce *Pii*-triggered immunity, suggesting a role for OsExo70 as a decoy or helper in *Pii/AVR-Pii* interactions [59,83,84].

The fourth type of *R*–*Avr* interaction is the indirect interaction between *Piz-t* and *AvrPiz-t*. This interaction may represent a classic example where a single *AvrPiz-t* interacts with different rice proteins to suppress immunity. However, the broad-spectrum R protein recognizes these proteins to restore or enhance immune response [85]. The product of *AvrPiz-t* is a secretory similar to other common *Avr* genes, which is composed of Cys62- Cys75 disulfide-bonded six-strand β-sheets [25]. The structure of *AvrPiz-t* and a similar gene, *ToXB*, have been determined by Nuclear Magnetic Resonance. A single point mutation in any of the cysteine residues diminishes the avirulence of *AvrPiz-t* [86]. Using yeast two-hybrid analysis, several AvrPiz-t interacting proteins (APIPs) in rice were identified, and AvrPiz-t was found to directly interact with APIP4 (encoding a bowman-birk trypsin inhibitor protein), APIP5 (encoding a bZIP transcription factor), APIP6 (encoding a RING E3 ubiquitin ligase), APIP10 (encoding a RING-type E3 ligase) and OsAKT1 (encoding a plasma-membrane-localized K^+^ channel protein) to disturb the rice PTI response [25]. For instance, APIP4 exhibits trypsin inhibitor activity and is required for rice innate immunity, while, upon infection, the *M. oryzae* effector *AvrPiz-t* interacted with APIP4 and suppressed APIP4 trypsin inhibitor activity. Interestingly, Piz-t can interact with APIP4 and enhance its accumulation and activity, which leads to resistance against virulence strains [44]. AvrPiz-t may target APIP10 for degradation, but, in return, APIP10 may ubiquitinate AvrPiz-t, causing its degradation [87]. This results in the silencing of APIP10 in the non-*Piz-t* background, which compromises the basal defense against *M. oryzae*, while silencing it in the *Piz-t* background causes cell death and enhances resistance. Most recently, APIP10 was found to directly interact with two rice transcription factors, VASCULAR PLANT ONE-ZINC FINGER 1 (OsVOZ1) and OsVOZ2, which is required for defense response. Notably, both OsVOZ1 and OsVOZ2 were found to interact with Piz-t and stabilize its transcription and accumulation, indicating the two proteins positively contribute to Pi-zt-mediated immunity [66]. During the necrotrophic stage of *M. oryzae* in rice, APIP5 negatively regulated necrosis or cell death, while Avrpi-zt could interact with APIP5 and suppress its transcriptional activity and protein accumulation. At the same time, Pi-zt interacts with APIP5, which may stabilize each other for either preventing necrosis mediated by APIP5 or enhancing immunity mediated by Piz-t [85]. Most recently, APIP5 directly interacts with OsWAK5 and CYP72A1, which play roles in ROS production and defense compound accumulation, respectively [44]. These studies also suggest that being a broad-spectrum *R* gene, the type of resistance or immune response depends not only on the type of APIPs but also the genetic background of rice in which *Piz-t* exists. For instance, PTI is suppressed by *AVrPiz-t* in *Piz-t*-lacking Nipponbare rice while stabilizing *Piz-t* in the *Piz-t* background when infected by *M. oryzae* [87].

**Table 2 ijms-24-04985-t002:** List of all cloned *Avr* genes.

*Avr* Genes	Protein Size (aa)	Cognate Cloned *R* Genes	References
*PWL1*	147	Unknown	[88]
*PWL2*	145	Unknown	[89]
*AVR1-CO39*	89	*Pi-CO39*	[67,90]
*AVR-Pita*	224	*Pi-ta*	[74]
*ACE1*	4035	*Pi33* (Un-clnoed)	[91]
*AVR-Pia*	85	*Pia*	[84,92]
*AVR-Pii*	70	*Pii*	[84]
*AVR-Pik/km/kp (AVR-Pikh)*	113; 5 Alleles (A-E)	*Pik/Pik-m/Pik-p*, *Pik-h*	[78,84]
*AvrPiz-t*	108	*Piz-t*	[25]
*AVR-Pi9*	91	*Pi9*	[93]
*AVRPib*	75	*Pib*	[44]
*AVR-Pi54*	153	*Pi54*	[72]
*MoHTR1*	Unknown	Unknown	[94]
*MoHTR2*	Unknown	Unknown	[94]

## 4. Current Status and Problems of *R* and *qR* Genes Utilized in Developing Broad-Spectrum Resistance Variety

Nowadays, marker-assisted selection (MAS) has been widely and successfully used to develop rice varieties with blast *R* genes [95,96]. The list of *R* genes and their corresponding molecular markers have been applied by different researchers to develop rice blast-resistant varieties (Table 3). By MAS, Feng et al. (2022) developed a new cultivar ‘Yangnonggeng 3091’ with the introgression of *Pigm* [97]. The new variety shows excellent blast resistance, tested by 184 isolates collected from rice growing regions in the lower region of the Yangtze River, as well as good performance on both grain yield and quality. However, due to the fact that large-scale deployment of the same *R* genes in rice cultivars brings uniformity, this ultimately causes the corresponding *M. oryzae* strain to undergo mutations. Such mutations lead to the emergence of a new virulent resistance-breaking strain, often resulting in pandemics owing to the high specificity of the blast resistance genes [98]. Therefore, although race-specific resistance is robust and effective against a particular pathotype, it is not durable. Regarding these concerns, besides utilizing broad-spectrum *R* genes, it is important to identify the distribution frequency of these known *R* genes in varieties in a particular area and then design varieties by pyramiding appropriate *R* genes. The application of combinations of different genes and pyramiding them in a single rice cultivar for developing broad-spectrum durable resistance is the most desired strategy. Several rice blast-resistant lines have been created using a pyramiding strategy with the three *R* genes *Pi2*, *Pi46* and *Pita* [96]. Using careful phenotyping followed by tagging with molecular markers, groups of researchers have characterized diverse rice germplasm for the distribution of different rice blast *R* genes in cultivated rice varieties. Xiao et al. (2018) have characterized rice varieties in the Heilongjiang Province of China and found that the distribution frequencies (DFs) of *Pi-ta*, *Pi5* and *Pib* are higher than those of other genes, reaching 31.37%, 29.41% and 18.62%, followed by *Pi2*, *Pi-d2* and *Pi-d3* with DFs of 9.80%, 1.96% and 1.96%, respectively [99]. In another study, Li et al. (2019) identified the existence of *Pi54*, *Pi5*, *Pi-ta*, *Pib* and *Pikm* but not *Pi9* in the core rice germplasm in Ningxia Province, China [100]. In a contemporary study performed in Guizhou Province, China, Ma et al. (2018) identified relatively high DFs of *Pi5* and *Pi54* in local varieties, at 32.35% and 30.86%, respectively; while the DFs of *Pi9* and *Pi2* were relatively lower, at 2.56% and 2.47%, respectively [101]. Wang et al. (2022) genotyped 195 rice varieties in Jiangsu province using diagnostic markers of 14 known *R* genes and found that most varieties in Jiangsu province carried two to five known *R* genes, and none of them contained *Pigm* [66]. The distribution frequencies of *Pib*, *Pita* and *Pikh* were relatively high and all exceeded 45% in the varieties tested; the remaining 10 genes were under 30%. Notably, after combining with the phenotype of panicle blast resistance, they further found that only three (*Pita*, *Pia* and *Pi3/5/i*) of these gene loci showed a significant contribution to panicle blast resistance and observed significantly positive interaction effects on resistance between *Pita* and either of the other two gene loci. This indicates that *Pita* and *Pia* or *Pi3/5/I* are appropriate gene combinations for developing resistant varieties against blast disease, at least in Jiangsu province. Using near-isogenic lines (NILs), several previous studies have also identified that complex interactions exist among different *R* genes [78,99]. This demonstrates that identifying appropriate *R* genes for pyramiding is important for breeding programs in a certain region. However, very little effort has been carried out in this important field so far. 

Quantitative resistance or partial resistance permits the development of lesions but halts lesion expansion and spore formation, thereby slowing down the infection and conferring sustained or prolonged resistance. Such resistance is durable and broad-spectrum due to low selection pressure on causative pathogens, which is less likely to mutate its population and minimizes the chances of an emergence of new resistance-breaking strains [39]. Therefore, current breeding programs are aimed to develop elite rice germplasm by deploying durable partial resistance to manage rice blast disease. To date, over 350 QTLs have been identified in different rice germplasm and several of these large-effect QTLs have been deployed to curb rice blast disease. Recent studies on the genetic analysis of partial blast resistance have been documented, mainly targeting QTLs such as *Pb1*, *pi21*, *Pi34*, *Pi35* and *Pi39* with the help of molecular markers tightly linked to these QTLs [19,24]. MAS has greatly facilitated rice breeders to characterize and select rice lines possessing rice blast QTLs of interest in the past decade. The deployment of a single partial resistance might not have been proved to effectively control rice blast, instead stacking multiple QTLs in a single rice line has contributed to durable and broad-spectrum resistance. However, details related to pyramid-suitable QTLs have been rarely documented. In one such study, Fukuoka et al. (2012) documented the pyramiding of three major rice blast QTLs, i.e., *qBR4-2a*, *qBR4-2b* and *qBR4-2c*, and significantly reduced the blast lesion area [102]. In total, compared with the cloned *R* genes, the small number of *qR* genes isolated so far is the critical problem that limits the breeding utilization of *qR* genes or resistance QTLs in practice. 

## 5. Current Status and Problems of Avirulence Genes Utilized in Practice for Guiding Rice Breeding and Blast Management 

Based on the gene-for-gene concept, the distribution frequency of *Avr* genes in pathogen population and the corresponding *R* genes in rice varieties can be employed to indirectly predict the epidemic severity of rice blast in a specific region as well as guide breeders to select appropriate *R* genes in breeding programs. For instance, Selisana et al. (2017) investigated the resistance spectrum of *Avr* genes in different strains of *M. oryzae* and showed how such a resistance spectrum was used to estimate the resistance efficiency of various rice cultivars [103]. The resistance frequency of cognate *R* genes in different rice lines perfectly matched the frequency of *Avr* genes. By applying genetic and molecular marker analysis, they identified additional *R* genes, most likely alleles of *Pi19* in rice cultivars. This study demonstrated that early diagnosis based on *Avr* genes can precisely predict the specificity and effectiveness of resistance conferred by different *R* genes in various rice cultivars, which in turn is crucial for predicting and managing rice blast epidemics (Table 4). Using the specific and diagnostic markers for each of the 10 *Avr* genes, one can predict the distribution of strains/isolates containing specific avirulence genes in a population [103].

Despite the cloned *Avr* genes and diagnostic markers developed for their identification, there are challenges to utilizing the given information on avirulence genes in practice. Possible reasons are as follows. First, the number of characterized avirulence genes is too few to fully represent the characteristics of the pathogen population. So far, only 14 *Avr* genes have been cloned and well characterized, contrary to diverse populations of *M. oryzae* strains (Table 2). In order to ensure the best utilization of *Avr* genes in practice, there is a pressing need to clone and characterize as many *Avr* genes as possible. Second, there are too many isolates or strains in field condition which need a large-scale collection of isolates [104], resulting in inconvenience and needing a highly efficient and low-cost genotyping method. Different strains of *M. oryzae* are routinely screened by different genotyping methods such as RAPD, SSR, SNP and FFLP. Third, high sequence variations in *Avr* genes may affect the value of the result in practice [105]. The high sequence variations are due to extensive gain or loss of genes via point mutation, transposition and translocation [106]. Such extensive evolutionary changes in the *M. oryzae* genome sometimes inactivate existing *Avr* genes while giving rise to new virulent isolates, thus complicating the practical significance of *Avr* genes in breeding programs. Fourth, due to the fact that the combination of some *R* genes could further broaden the resistance spectrum; however, the mechanism remains unclear, which means that the gene-for-gene relationship alone may not be enough to manage blast disease, including the utilization of appropriate *R* genes.

The key challenge to controlling rice blast is the constantly changing population of rice blast fungus and the emergence of new virulent strains. Thus, the best management strategy to detect constantly evolving *Avr* genes of rice blast fungus is to have an efficient surveillance system at hand. Thus, an effective surveillance system is necessitated to monitor emerging novel virulent strains of *M. oryzae.* The advent of Next Generation Sequencing (NGS) and molecular breeding tools means that rice breeders are now able to design more robust methods of rice blast surveillance and control. For example, Mutiga et al. (2021) in Africa recently proposed a robust pathogenomic-based rice blast surveillance system known as the Mobile and Real-Time Plant Disease (MARPLE) system [107]. In this system, infected rice leaf tissue is collected to identify the characteristics, genomic signatures and genetic shift in the pathogen population in the field. In the next step, DNA is isolated from infected leaves followed by enrichment of the DNA with putative avirulence genes via multiplex PCR prior to targeted genome sequencing by oxford nanopore sequencer. The sequencing data are subsequently analyzed to predict avirulence gene evolution and acquired fungicide resistance. Based on the analyzed data, rice breeders make precise and timely decisions to breed and deploy virulent-specific rice cultivars to curb rice blast disease. 

## 6. Future Perspectives

### 6.1. Mining New R Genes and Investigating the Interactions among R Genes Deeply

Due to the fact that it is impossible to stop the evolution of blast fungus and host rice, it is important to unceasingly mine new *R* genes. Although pyramiding *R* genes is undoubtedly a good approach to developing a broad-spectrum resistance variety, complex interactions do exist among known *R* genes. The effective deployment and utilization of an effective combination of different *R* genes in breeding against rice blast has been found to be challenging owing to the abundance of *R* genes in rice and their complex interaction mechanisms [108]. Therefore, it is of critical importance to find those *R* gene combinations with a positive interaction for increasing the resistance spectrum and to elucidate their molecular mechanism. Some studies have shown that the interaction between *R* genes during pyramiding is linked with the number of *R* genes being pyramided. Thus, combining more *R* genes in the same cultivar reflects higher resistance against *M. oryzae* and vice versa [100,106]. However, in more practical cases, increasing the number of *R* genes slows down the level of resistance of the cultivar owing to the linkage drag associated with multiple pyramided genes [93,99]. Thus, developing key positive interactions among *R* genes during pyramiding is attributed to understanding the interaction mechanism, screening different R combinations and finally deploying the right combination pattern of *R* genes in single rice cultivar. Xiao et al. (2016) have successfully pyramided *Pi-ta* and *Pi46* and broadened the resistance spectrum of the pyramided line as compared to monogenic lines [109]. Similarly, the perfect combination and interaction of *Pik/Piz* pairs in the rice variety ‘Jefferson’ have proved broad-spectrum resistance since 1997 [110,111]. It is suggested that more novel *R* genes should be mined, and their interaction mechanism should be investigated deeply for developing novel types of broad-spectrum resistance. 

### 6.2. Identifying More qR Genes and Evaluating Their Breeding Potential 

Due to the importance of *qR* genes in developing durable resistance variety, deeply mining *qR* genes is particularly important. Genome-wise association studies (GWAS) should be a promising strategy to widely identify *qR* genes or susceptible genes. This process is performed in three steps to identify more *qR* genes via GWAS. In the first step, different rice cultivars containing different *Pi* genes are subjected to artificial inoculation with different strains of *M. oryzae* that are prevalent in a specific cultivation area. This, in turn, helps to characterize different prevalent isolates of *M. oryzae*. In the second step, multiplex and robust genotyping methods are used to characterize both the resistant cultivars and their corresponding pathogens simultaneously. Most importantly, the abundant SNP markers available via a robust and multiplex genotyping system further assist in GWAS to identify novel genomic regions either associated with single *R* genes of race-specific resistance or *qR* genes conferring broad-spectrum resistance against *M. oryzae* isolates. The third phase is to combine the results of the above mentioned steps for gene mining via multiple crosses to develop a variety that confers broad-spectrum rice blast resistance [107]. In the assessment phase, the newly developed rice cultivars containing novel *qR* genes are evaluated for multiple years and at multiple locations to validate their durability and broad-spectrum resistance against different isolates of *M. oryzae*. 

### 6.3. Accelerating the Application of Molecular Technologies in Breeding Program

The application of molecular technologies such as MAS, genomic selection and genome editing is hoped to not only assist but also speed up breeding programs aimed at developing rice blast-resistant rice cultivars. Conventional breeding techniques are not only costly and labor intensive, but also need a long time to develop rice blast-resistant varieties. On the contrary, genomic selection assisted by molecular markers shortens the time required for cultivar development by selecting blast-resistant lines in earlier generations. Moreover, genomic selection followed by MAS potentiates the precise pyramiding of several candidate *R* genes into a single cultivar for developing durable and broad-spectrum rice blast resistance. 

Progress made in rice genomics and development has enabled rice breeders to clone more and more rice blast *R* and *M. oryzae Avr* genes. The successful cloning of *Pi2*, *Pi9* and *Pigm* opened new avenues to identify functional SNPs closely associated with resistant genotypes. Such functional SNPs can be used to develop robust Kompetitive Allele-Specific PCR (KASP) markers in abundance for marker-assisted rice blast resistance breeding programs [107]. In a similar study, Wang et al. (2019) sequenced and assembled a high-quality genome of ‘Tetep’, a broad-spectrum rice blast-resistant germplasm which is also the donor of the *Pi5* gene [112]. A total of 455 *NLRs* genes were predicted in the genome assembly ‘Tetep’ [112]. Molecular markers designed from these predicted *NLRs* have not only enabled rice breeders to select resistant rice cultivars, but have also assisted in introducing these *NLRs* to new breeding varieties for durable rice blast resistance. To grasp the practical potential of genomic selection and MAS, Feng et al. (2022) evaluated 162 accessions from the USA for their resistance to six rice blast isolates and found that genomic selection and MAS can be effectively used for rice blast resistance [97]. Xiao et al. (2019) utilized the potential of MAS to pyramid *Pi2*, *Pi46* and *Pita* and developed broad-spectrum rice blast-resistant lines [96]. Furthermore, Xiao et al. (2019) successfully introgressed broad-spectrum resistance gene *Pi2* into the genetic background of an elite Chinese rice cultivar ‘Feng39S’ through SNP array-based marker-assisted backcrossing coupled with genomic-based background selection, and the newly developed rice line ‘Feng39S’ with durable resistance was suggested to replace the original parent in developing the popular hybrid rice variety ‘Fengliangyou4’ [96].

Genome Editing Technologies (GETs) have emerged as key players in gene functional research. Among TEGs, CRISPR/Cas9 has been widely adapted by scientists as robust, technically less demanding and precise gene-editing tools. This technology is becoming the prime choice of gene editing tools by rice breeders. The function of several rice blast resistance genes has been validated recently. For example, Wang et al. (2016) selectively mutated the *OsERF922* gene in rice with CRISPR/Cas9 and found that the mutated line conferred higher resistance to *M. oryzae* than the wild type [85]. Similarly, the targeted mutation of the rice blast durable resistant *Ptr* gene via CRISPR/Cas9 rendered the gene susceptible, thus successfully validating gene function [21]. Most recently, in aid of CRISPR/Cas9, two studies reported the generation of rice lines with broad-spectrum resistance to blast variety by editing two genes, *Pi21* and *Bsr-d1*, and found that simultaneously editing the two genes had much stronger resistance than editing one of them [29]. These findings prove that simultaneous editing of numerous *S* genes is an effective method for creating novel rice cultivars with broad-spectrum resistance. In the near future, CRISPR/Cas9-mediated gene editing technology will be undoubtedly widely used to develop sustainable and durable resistance rice cultivars against blast disease.

### 6.4. Isolation of More Avr Genes and Accelerating the Elucidation of Interactions between Both R-Avr Pairs and Different Avr Genes

The available sequence genome of *M. oryzae* has led to the successful isolation of 14 *Avr* genes (Table 2). However, compared to the isolation of 38 *R* genes, the progress of the isolation of *Avr* genes from *M. oryzae* strains remains slower. One of the reasons is that the rice blast *Avr*-gene family is highly diversified, and the pathogen is capable of rapidly undergoing evolutionary changes in the form of retrotransposons, deletions, translocations and point mutations. Such drastic evolutionary changes might lead to a loss of avirulence [113]. Some classical *Avr* genes, such as *Avr-Pia*, *Avr-Pii* and *Avr-Pik*, were found completely absent in assembled rice blast fungal genomes [84]. This shows that presence/absence polymorphism is a driving evolutionary force of *Avr* genes. Such presence/absence polymorphism hampers the cloning of *Avr* genes as shown by failed amplifications of six *Avr* genes from different strains by different primer combinations [114]. Multiplex methods should be devised to accelerate the cloning of *Avr* genes from diverse strains of *M. oryzae* as well as to determine its sequence variance in the *M. oryzae* population. In addition, the interplay of *R-Avr* pairs is important for an efficient surveillance system and for deploying broad-spectrum resistance. Some of these interacting models have been shown to confer broad-spectrum resistance against rice blast [115]. However, the interaction of *R-Avr* and *R-R* gene pairs has been investigated in detail in the past few decades, and the detailed interaction mechanisms between most isolated *R* and *Avr* genes remain to be elucidated. In addition, attention should also be focused on investigating the interaction among different *Avr* genes in *M. oryzae* strains. In cereal powdery mildew, it has been shown that the interaction of *Avr* with the suppressor of the *Avr* gene could lead to the mechanism of recognition specificity [115,116,117]. This implies that complex interactions might exist not only in *R*-*Avr* pairs and *R*-*R* genes but also in *Avr*-*Avr* genes, which together affect the host phenotype, being either resistant or susceptible. Therefore, the isolation of more *Avr* genes and elucidating their interactions with both *R* genes and themselves are quite important for better managing rice blast. 

### 6.5. Development of New Type of Fungicide Based on the Pathogenic Mechanism

The extensive application of a wide array of modern fungicides not only dramatically reduced damage caused by *M. oryzae* strains but also greatly enhanced the quality and yield potential of global rice production. The frequent deployment of the same fungicide may lead to the emergence of novel pathogenic strains that further necessitate the discovery of novel compounds to curb rice blast resistance-breaking strains. Appressorium formation is required for the successful infection of *M. oryzae*, and melanin plays an important role in appressorium growth and penetration via the cuticle layer of the rice plant [116,117]. Thus, melanin biosynthesis inhibitors (MBIs) have been used to suppress appressorium growth and penetration via the cuticle layer. Broadly, two groups of MBIs fungicides, i.e., scytalone dehydratase (MBI-D) and poly-hydroxynaphthalene reductase (MBI-R), proved fatal against appressorium growth inside rice plants. The latter group, comprising mainly phthalide, pyroquilon and tricyclazole, witnessed no resistant pathogen emergence despite 30 years of longer duration widespread application. Few resistant mutants have been identified in laboratories in China without the emergence or isolation of any resistant strain in the field [118]. Most recently, He et al. (2020) found that *M. oryzae* utilizes a special class of enzymes called septin GTPases for the appressorium-mediated development of infection [119]. However, septin GTPases require very long-chain fatty acids (VLCFAs) for membrane-based septin assembly for infection progression. The chemical fungicides developed by the group contain inhibitors of septin biosynthesis, thereby affecting septin assembly and appressoria-based host membrane penetration. The advantage of the septin inhibitor-based novel class of fungicides is that these compounds do not only control rice blast but also confer broad-spectrum resistance against fungal pathogens of plants and animals without affecting the target host [119]. 

## 7. Conclusions

The development of a resistant variety is one of the most economical and efficient approaches to control rice blast, which requires an understanding of the mechanism of both rice resistance and pathogen pathogenicity. This review summarized the advances in the isolation of R and qR genes from the rice host and Avr genes from *M. oryzae* and that the interactions between the rice host and pathogen depend on R-Avr gene pairs. In addition, the problems of using these advances to manage the disease were discussed. Based on these advances and potential problems, five research focuses for the future were suggested, which are crucial for developing a broad-spectrum resistance variety.

## Figures and Tables

**Table 3 ijms-24-04985-t003:** List of molecular markers for MAS of rice blast.

Genes	Varieties	Markers	Marker Type	Chr. No	Tm (°C)	Size (bp)
*Pi1*	C101LAC	RM1233	SSR	11	55	155/170
*Pi2*	IRBLZ5-CA	Pi-2GM	Functional	6	55	2344
*Pi5*	IRBLz5-M	40N23r	InDel	9	55	480/700
*Pi9*	IRBL9-w	195R-1	STS	6	56	2000
*Pikm*	IBLKMTS	k2167	InDel	11	55	300/619
*Pib*	IRBLB-b	Pb28	SNP	2	60	388
*Pik*	IRBLZK-Ka	K39512	SNP	11	60	112
*Piz*	IRBLZ-Fu	Z56592	SNP	6	60	388
*Pik-h*	IRBLKH-k3	candidate	Functional	11	55	1500
*Pik-p*	IRBLZKP-k60	K3957	SNP	11	60	148
*Piz-t*	IRBLZT-t	Zt56591	SNP	6	60	257
*Pita*	IRBLTA-K1	Pita3	SNP	12	59	861

**Table 4 ijms-24-04985-t004:** List of molecular markers to identify *Avr* genes.

Primer Name	Tm (°C)	*Avr* Detected
AVR-Pi9 GF & GR	55	*AVR-Pi9*
AVR-Pii F & R	58	*AVR-Pii*
AVR1-CO39 F & R	57	*AVR1-CO39*
AVR Piz-t F & R	56	*AVRPiz-t*
AVR-Pik F & R	57	*AVR Pik*
Avr-Pia F & R	58	*AVR-Pia*
AVR-Pita F & R	58	*AVR-Pita*
RM224 F & R	58	*Pikp*
RM27920 F & R	58	*Pi19*
RM27947 F & R	58	*Pi19*

## Data Availability

Not applicable.

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
