# Peer review of "Approaches to Reduce Rice Blast Disease Using Knowledge from Host Resistance and Pathogen Pathogenicity"

_ijms, 2023, doi:10.3390/ijms24054985_

Round 1
Reviewer 1 Report
The manuscript is impressively well written; I had only minor spelling and grammar issues. I have attached highlighted text with comments in-line in the attached pdf.
The only significant issue I observed is that there is inconsistent use of italics to declare gene names throughout the paper. In particular, I became confused in particularily dense sections where genes were referred to in mixed fonts. This may be more due to a convention of writing I am not versed in.
Additionally, cultivar names lacked punctuation inconsistently (e.g. early in the paper you used backticks). I would recommend just making punctuation more consistent in the final version.
Other than these small issues, it was very well done!

Author Response
The manuscript is impressively well written; I had only minor spelling and grammar issues. I have attached highlighted text with comments in-line in the attached pdf.
Response: Thanks for the recognition on the manuscript and excellent suggestion provided in the attached pdf. In the revised version, we have carefully modified these parts according to the accordingly, and all revisions were recorded in ‘shown maker-up’ in Word.
The only significant issue I observed is that there is inconsistent use of italics to declare gene names throughout the paper. In particular, I became confused in particularly dense sections where genes were referred to in mixed fonts. This may be more due to a convention of writing I am not versed in.
Response: Thanks for the comments. In the revised we have carefully revise these throughout the manuscript.
Additionally, cultivar names lacked punctuation inconsistently (e.g., early in the paper you used backticks). I would recommend just making punctuation more consistent in the final version.
Response: Thanks for the suggestion, we have corrected these and used a consistent punctuation for this in the revised version.
Other than these small issues, it was very well done!
Response: Thank you once again.

Reviewer 2 Report
Plagiarism is higher.
The reference 11, 12 on ill effect of fungicide does not denote about the specific uses directly coming from pesticides used against rice blast, rather the cited papers are general in nature.
The NB LRR that are receptors to effectors does not explain, whether they are bacteria specific or broad to all pathogens.
The hemi biotrophy nature of M. oryzae, there is no mention about the timing of it when the biotrophic nature ends and necrotrophic cycle begins.
Some sentences are not complete as seen in pg 7 “Developed several broad-spectrum resistant rice blast lines by pyramided three R genes Pi2, Pi46 and Pita’
Reference misplacing after completed sentences and at the beginning of the next sentences are also seen
Sometimes, authors names are not mentioned as in pg 7, ref 60, 62, 66, 67, 68 (pg 8), 69 , 73, 79 and many other places.
Modified Fig 1, Do the Authors have permission in doing so…..?
The subtitle 6.4 may be put with a meaningful line.
M. oryzae is also mentioned as M. grisea in few places, please correct.
The developmental cues on how M. oryzae infects rice may be discussed with plausible approach towards developing resistance.
A final Concluding paragraph could be valuable.
Author Response
Plagiarism is higher.
Response: Thanks for this scientific comment. Due to a hot research field, there has many review papers, and we have to fully avoid the similarity with the other publications. Certainly, in the revised version, we have tried our best to reduce this to a great extent.
The reference 11, 12 on ill effect of fungicide does not denote about the specific uses directly coming from pesticides used against rice blast, rather the cited papers are general in nature.
Response: Thank you very much for your deep concern. In fact, in this sentence, we just wanted to show others’ negative effects of fungicides and not limited to the fungicides against rice blast. Since this is a general phenomenon in practice, the common publications were cited accordingly.
The NB LRR that are receptors to effectors does not explain, whether they are bacteria specific or broad to all pathogens.
Response: Thanks for the comment. In the revised version, we have added a short explanation for this, and the NB-LRR receptors are very conserved in plants against various pathogens.
The hemi biotrophy nature of M. oryzae, there is no mention about the timing of it when the biotrophic nature ends and necrotrophic cycle begins.
Response: In infection, the time of M. oryaze switch from biotrophy to necrotrophic has been added in the introduction section in revised version.
Some sentences are not complete as seen in pg 7 “Developed several broad-spectrum resistant rice blast lines by pyramided three R genes Pi2, Pi46 and Pita’
Response: Thank you so much, and as of page 7, the sentences have been polished accordingly.
Reference misplacing after completed sentences and at the beginning of the next sentences are also seen. Sometimes, authors names are not mentioned as in pg 7, ref 60, 62, 66, 67, 68 (pg 8), 69, 73, 79 and many other places.
Response: Thanks for you pointing these errors. In the revised version, we have carefully edited these errors throughout the manuscript.
Modified Fig 1, Do the Authors have permission in doing so…..?
Response: Thanks for this friendly comment. In order to avoid the potential troubles, we decided to delete this Fig in the revised version, and just keep the description and citation in the text file.
The subtitle 6.4 may be put with a meaningful line.
Response: Thanks. We have changed it to “Isolation of more Avr genes and accelerating the elucidation of interactions between both R-Avr pairs and different Avr genes”.
- oryzae is also mentioned as M. grisea in few places, please correct.
Response: Thanks for pointing this mistake. In the revised version, we have corrected it accordingly.
The developmental cues on how M. oryzae infects rice may be discussed with plausible approach towards developing resistance. A final Concluding paragraph could be valuable.
Response: Thanks for the comments. In the revised version, we have added a brief introduction about M. oryzae infection in the introduction section and a conclusion paragraph in the revised version.

Reviewer 3 Report
Magnaporthe oryzae causes the most serious disease in rice fields worldwide. In this review, the authors highlight an important topic on the discovery of Avr and resistant genes as well as the current progress on rice breeding and rice blast management.
In general, this manuscript covers the current findings and mechanisms of some Avr genes. Here are some comments on reorganization and edits that could help improve the manuscript.
Overall, the authors should read through the manuscript and revise it for general typos, grammar, formatting, and sentence structures.
Section 1: Magnaporthe oryzae lifestyle and the description of the infection structures (such as the appressorium) need to be included in the introduction section. Additionally, a brief summary of the mechanisms of R and Avr gene interactions should be included in this section, highlighting the importance of this interaction for immune responses.
Section 2: The sentence beginning with “Different studies have demonstrated that resistance mechanism against or Magnaporthe oryzae is complex…” needs to be edited for clarity. Ensure that organism names are abbreviated following the introduction of the full name in later portions of the text. Additionally, the authors should revise the manuscript to ensure italics are used properly.
Section 3: Authors should include the new research findings on the molecular mechanisms employed by Avr-pita effector in the host. Here is the reference: Han, J., Wang, X., Wang, F. et al. The Fungal Effector Avr-Pita Suppresses Innate Immunity by Increasing COX Activity in Rice Mitochondria. Rice 14, 12 (2021).
Section 3: “M. oryzae spreads in rice biotrophically during early infection and then switches to necrotrophic phase, which is considered as hemibiotrophic pathogen.” This sentence should be moved to the introduction, where the authors expand on M. oryzae life cycle
Section 4: Many references in this section begin at the start of the sentences. Please ensure that this formatting is accepted for the journal.
Section 5: In this section, the authors list four limitations in studying Avr genes for disease control. Specifically, their fourth limitation suggests that gene-for-gene relationships alone will not be enough to manage blast disease. The authors should include the importance to study these interactions despite these limitations.
Section 6.2: GWAS should be introduced in section 6.2 instead of in section 6.4.
Section 6.5: “M. grisea develops a structure known as melanized appressoria that is used by the pathogen to penetrate the hard cuticle layer of rice plant.” This sentence should be in the introduction section and it requires a reference paper.
The authors should consider adding a concluding paragraph at the end of the manuscript highlighting the most important areas of research that could potentially lead to effective disease management control.
Author Response
Magnaporthe oryzae causes the most serious disease in rice fields worldwide. In this review, the authors highlight an important topic on the discovery of Avr and resistant genes as well as the current progress on rice breeding and rice blast management.
In general, this manuscript covers the current findings and mechanisms of some Avr genes. Here are some comments on reorganization and edits that could help improve the manuscript.
Overall, the authors should read through the manuscript and revise it for general typos, grammar, formatting, and sentence structures.
Response: Thanks for the comments. We have carefully revised the manuscript for general typos, grammar, formatting, and sentence structures, and feel that the revised version is acceptable.
Section 1: Magnaporthe oryzae lifestyle and the description of the infection structures (such as the appressorium) need to be included in the introduction section. Additionally, a brief summary of the mechanisms of R and Avr gene interactions should be included in this section, highlighting the importance of this interaction for immune responses.
Response: Thank you very much for these comments. We have added the description of the infection structures in the introduction section as well as a brief summary of the mechanism of R and Avr gene interactions.
Section 2: The sentence beginning with “Different studies have demonstrated that resistance mechanism against or Magnaporthe oryzae is complex…” needs to be edited for clarity. Ensure that organism names are abbreviated following the introduction of the full name in later portions of the text. Additionally, the authors should revise the manuscript to ensure italics are used properly.
Response: Thanks for you pointing these errors. In the revised version, the sentence has been changed to “Different studies have demonstrated that rice resistance to M. oryzae is complex and often involves the interaction of various qR and R genes in a synergistic approach”, and we have also carefully edited these mistakes throughout the manuscript.
Section 3: Authors should include the new research findings on the molecular mechanisms employed by Avr-pita effector in the host. Here is the reference: Han, J., Wang, X., Wang, F. et al. The Fungal Effector Avr-Pita Suppresses Innate Immunity by Increasing COX Activity in Rice Mitochondria. Rice 14, 12 (2021).
Response: Thanks for the suggestion. The new research findings has been added to the manuscript, and thanks again.
Section 3: “M. oryzae spreads in rice biotrophically during early infection and then switches to necrotrophic phase, which is considered as hemibiotrophic pathogen.” This sentence should be moved to the introduction, where the authors expand on M. oryzae life cycle
Response: Yes, the sentence has already been moved to the introduction section.
Section 4: Many references in this section begin at the start of the sentences. Please ensure that this formatting is accepted for the journal.
Response: Sorry for these mistakes. In the revised version, the references have already been corrected throughout the manuscript.
Section 5: In this section, the authors list four limitations in studying Avr genes for disease control. Specifically, their fourth limitation suggests that gene-for-gene relationships alone will not be enough to manage blast disease. The authors should include the importance to study these interactions despite these limitations.
Response: Thanks for the suggestion. We have emphasized the importance on studying the interactions from R genes, R-Avr pairs and Avr genes in the revised version.
Section 6.2: GWAS should be introduced in section 6.2 instead of in section 6.4.
Response: Thanks for the suggestion. The full name of GWAS has been moved to section 6.2 in the revised version.
Section 6.5: “M. grisea develops a structure known as melanized appressoria that is used by the pathogen to penetrate the hard cuticle layer of rice plant.” This sentence should be in the introduction section and it requires a reference paper.
Response: We have added this description in the introduction section
The authors should consider adding a concluding paragraph at the end of the manuscript highlighting the most important areas of research that could potentially lead to effective disease management control.
Response: Yes, we have added a conclusion paragraph at the end of the manuscript.

Round 2
Reviewer 2 Report
the plagiarism remains 22% which must be reworded to reduce it to less than 10%.